# SymLight: Exploring Interpretable and Deployable Symbolic Policies for Traffic Signal Control

## Abstract

Deep Reinforcement Learning have achieved significant success in automatically devising effective traffic signal control (TSC) policies. Neural policies, however, tend to be over-parameterized and non-transparent, hindering their interpretability and deployability on resource-limited edge devices. This work presents SymLight, a priority function search framework based on Monte Carlo Tree Search (MCTS) for discovering inherently interpretable and deployable symbolic priority functions to serve as the TSC policies. The priority function, in particular, accepts traffic features as input and then outputs a priority for each traffic signal phase, which subsequently directs the phase transition. For effective search, we propose a concise yet expressive priority function representation. This helps mitigate the combinatorial explosion of the action space in MCTS. Additionally, a probabilistic structural rollout strategy is introduced to leverage structural patterns from previously discovered high-quality priority functions, guiding the rollout process. Our experiments on real-world datasets demonstrate SymLight's superior performance across a range of baselines. A key advantage is SymLight's ability to produce interpretable and deployable TSC policies while maintaining excellent performance. Our codes will be released upon acceptance.

## 1 Introduction

Deep Reinforcement Learning (DRL) has emerged as a powerful approach to Adaptive Traffic Signal Control (ATSC), enabling automatic learning of traffic signal policies through interactions with dynamic traffic environments. Despite promising research results, the real-world adoption of DRL-based ATSC remains surprisingly low. In North America, fewer than 5% of signalized intersections currently utilize ATSC systems (Berman, 2021), with even lower adoption rates in developing regions. In practice, traditional fixed-time or actuated strategies continue to dominate urban traffic management (Xing et al., 2022). This disconnection between research advancements and practical deployment can be attributed to several critical limitations inherent in current DRL-based approaches. These limitations hinder both the trustworthiness and feasibility of deploying DRL in real-world traffic environments.

First, DRL policies often lack interpretability (Gu et al., 2024). Since DRL relies heavily on neural networks, the resulting policies are usually black-box models. This opacity poses significant concerns: traffic engineers struggle to validate or troubleshoot learned behaviors; authorities find it difficult to justify decisions and establish public trust; and drivers lack insight into how signal changes are determined—reducing compliance and potentially increasing frustration or unsafe driving behaviors.

Second, beyond interpretability, a practical TSC policy should also be readily deployable (Xing et al., 2022; Chauhan et al., 2020) on existing infrastructure, which are typically low-cost edge devices with significant constraints on computation and memory. Deploying complex neural policies on such hardware leads to significant latency in decision-making and violation of the memory constraints. To make the policies deployable, necessary model compression (e.g., quantization) is commonly adopted (Han et al., 2015). However, this often greatly degrades their performance.

Third, there is often a misalignment between training objectives and real-world evaluation metrics. In practice, DRL requires a well-defined reward function to guide policy learning. However, there is often a fundamental mismatch between the reward used in training and the actual performance indicators mandated by traffic authorities. While most DRL approaches optimize some local surrogate metrics (such as the intersection queue length (Zheng et al., 2019b) and intersection pressure (Wei et al., 2019a)), public agencies typically evaluate traffic control systems based on some system-level objectives such as total throughput, average travel time, or environmental impact (e.g., emissions reduction). This misalignment between the training reward and the deployment objectives will create a gap between what is optimized during training and what is desired in deployment, leading to potential real-world unsatisfactory performance.

To address these challenges, we propose SymLight, a TSC framework that employs explicit symbolic expressions as its policy representation. Specifically, at each intersection, the TSC policy in SymLight is formulated as a learnable symbolic priority function, which computes a priority score for each available phase based on the current traffic conditions. The theoretical motivation for this phase priority control is rooted in MaxPressure control principle(Varaiya, 2013), the priority function in this work serves as a continuous-valued, phase-level, and feature-grounded abstraction that directly maps real-time traffic features to a unified priority score. Built on this decision-making framework, SymLight utilizes Monte Carlo Tree Search (MCTS) to thoroughly explore the discrete space of expressions, aiming to identify high-performing priority functions. Importantly, SymLight allows global, system-level objectives to be directly used as the reward signal during training, thereby avoiding common reward-objective mismatches and possible credit assignment challenges. Through these design choices, SymLight facilitates the discovery of TSC policies that are not only effective and interpretable, but also lightweight enough to be deployed on existing traffic signal hardware. The main contributions of this paper can be summarized as follows:

1. We propose a new representation for interpretable and deployable TSC policies. The TSC policy is represented by a symbolic priority function that takes traffic features as input and outputs a priority score for each traffic movement, which are then aggregated into the priority scores of the phases. To automatically discover effective priority functions, we introduce SymLight, a priority function search framework based on MCTS operating within a heuristic search space.

2. To address the combinatorial explosion issue in the action space in MCTS, we design a concise yet expressive encoding of the priority function that enables efficient search. To further enhance the representation, we introduce two traffic lane features that comprehensively capture critical lane-level traffic dynamics. Building on this foundation, we develop a probabilistic structural rollout strategy that exploits structural patterns identified in previously discovered high-quality priority functions to guide the rollout process more effectively.

3. Extensive experiments on real-world datasets demonstrate that SymLight outperforms a diverse set of baselines. Ablation studies further confirm the effectiveness of each individual design within SymLight. Notably, SymLight is capable of generating human-understandable, deployable, and non-parametric TSC policies, without compromising its performance.

## 2 BACKGROUND

### 2.1 PRELIMINARIES

In this section, we summarize relevant definitions for TSC.

**Traffic network** A traffic network, also referred to as a road network, is composed of various interconnected intersections and roads. Fundamentally, a road network can be represented as a directed graph, $G = (V, E)$, where $V$ denotes the set of nodes (intersections), and $E$ comprises the set of unidirectional links (roads).

**Road** A road $\mathcal{R}$ is typically composed of multiple lanes, which can include various types such as left-turn, go-straight, and right-turn lanes.

**Lane** A lane $l$ is a section of a road designated for use by a single stream of vehicles.

**Intersection** An intersection marks a point where multiple roads converge. It can therefore be conceptualized as a graph node featuring several incoming and outgoing links.

**Traffic movement** The act of vehicles traveling across an intersection, from an incoming lane to an outgoing lane, is termed a traffic movement.

**Traffic signal phase** A traffic signal phase in this work refers to a green light arrangement during which a particular set of traffic movements is allowed to proceed.

**Multi-intersection traffic signal control** At each time step within the analysis period, the algorithm is tasked with selecting the best traffic signal phase for every intersection, guided by real-time traffic conditions.

## 2.2 RELATED WORK

Traditional methods in the transportation field for solving TSC broadly fall into two categories: cycle-based methods (Li & Sun, 2018; Tung et al., 2014; Jia et al., 2019) and heuristic methods (Li & Jabari, 2019; Varaiya, 2013). Cycle-based techniques, which typically depend on predefined repeating patterns (Renfrew & Yu, 2012), inherently struggle with flexibility in complex traffic situations. Meanwhile, heuristic methods, employing manually designed rules to manage traffic signals, are heavily reliant on expert knowledge and lack the capacity to adaptively learn from real-world traffic data. These traditional methods, despite their strong interpretability, frequently struggle to perform effectively in complex and highly dynamic scenarios due to their inherent limitations (Wang et al., 2024).

In contrast to traditional methods, DRL-based methods can automatically acquire TSC policies (Wei et al., 2019a; Zhang et al., 2022; Liao et al., 2025b). They accomplish this through trial-and-error interactions with intersections, utilizing system feedback and eliminating the need for prior environmental knowledge (Zheng et al., 2019b). To illustrate, FRAP (Zheng et al., 2019a) proposes a novel network architecture centered on phase competition and relationships. This allows it to proficiently manage imbalanced traffic and guarantee symmetry invariance. CoLight (Wei et al., 2019b), on the other hand, utilizes graph attention networks to model inter-intersection communication, thereby facilitating cooperation among them. Furthermore, MPLight (Chen et al., 2020) adopts FRAP's neural network foundation to achieve decentralized city-level traffic signal control through parameter sharing. However, while these methods largely concentrate on achieving high performance, they tend to overlook other important considerations, including their interpretability and deployability. MCTS has also been previously applied to TSC, notably within the MCTS-IO (Qi & Hu, 2019). This method seeks the optimal phase sequence by explicitly modeling complex traffic dynamics (e.g., channelized section spillover). However, MCTS-IO relies on detailed, model-dependent traffic flow to derive its phase sequences. This dependency limits its transparency and deployability in real-world settings. Furthermore, as the method relies on online decision optimization, it is more time-consuming compared to offline trained policies.

The advent of Large Language Models (LLMs) has unveiled a fresh opportunity for their utility in TSC problems (Chen et al., 2025; Movahedi & Choi, 2025). A new trend sees researchers employing advanced LLMs as TSC agents, allowing for human-like and interpretable TSC decisions (Lai et al., 2023b; Wang et al., 2024). In contrast to DRL-based methods, LLM-based methods distinguishes itself by not only developing highly effective control policies but also by providing explicit, detailed justifications for every decision made (Lai et al., 2023a). However, LLM-based methods also face several limitations when applied to TSC. First, their decision-making speed could not meet strict real-time requirements due to either LLM inference speed or possible network latency. Second, deploying LLMs often demands significant computational resources, making them less practical for edge or low-power environments. Third, the factual accuracy and consistency of their reasoning can vary (Huang et al., 2025), which may lead to unreliable or inconsistent control decisions.

Recently, some researchers begin to explore practical TSC methods for real-world deployment. Recent works such as RobustLight (Li et al.) and FuzzyLight (Li et al., 2025) have made impressive progress in enhancing the robustness of TSC policies under perception noise and uncertain traffic conditions. Their innovative approaches demonstrate that TSC systems can maintain high performance even in extremely challenging real-world scenarios. In addition to this, designing lightweight network architectures suitable for edge devices has been a focus of recent work. For instance, Tiny-

Light (Xing et al., 2022) reduces large super-graphs into more compact networks through pruning. EcoLight (Chauhan et al., 2020) develops a lightweight network and then uses empirical thresholds to approximate it. While these lightweight neural policies offer a degree of deployability, they still exhibit non-transparent characteristics or necessitate a trade-off between performance and model size (Zhu et al., 2023). Recent advances (Young et al., 2019; Anderson et al., 2020) in neurosymbolic reinforcement learning demonstrate the potential of combining symbolic reasoning with learning to obtain interpretable policies in other domains (Paleja et al., 2022; Mundhenk et al., 2021). These methods, however, focus on tasks such as image generation and formally verified exploration, which are fundamentally different from TSC. Motivated by this direction, researchers have also begun to explore transparent and verifiable TSC policies. Instead of relying on opaque neural networks for TSC policies, some researchers also start to explore explicit TSC policy methods. GPLight (Liao et al., 2024) and GPLight+ (Liao et al., 2025a) use genetic programming to automatically evolve interpretable and deployable tree-shaped TSC policies. Nonetheless, a limitation of these methods is their reliance on homogeneous traffic phase settings, which are uncommon in real-world traffic scenarios. DRSQ (Ault et al., 2019) employs polynomial functions for TSC policy representation, $\pi$-Light (Gu et al., 2024) leverages learnable programs for the same goal. While successful in providing interpretable TSC, these methods are constrained by the inherent inflexibility of their policy representations, hindering the searching for more effective TSC policies. To address the above limitations, SymLight leverages a policy representation that is both concise and expressive to discover more effective TSC policies via MCTS.

## 3 SYMLIGHT METHOD

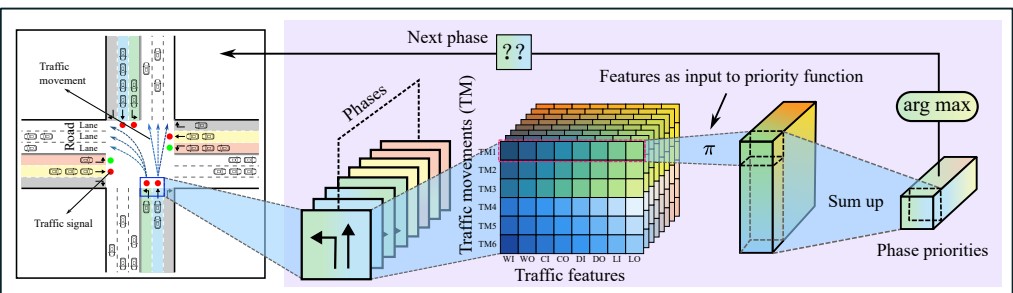

Figure 1: An illustration of a single phase decision at an intersection, showing the role of the priority function in SymLight. The priority function takes lane-level traffic features as input to determine the priority of each phase.

### 3.1 OVERALL FRAMEWORK

In SymLight, a policy is represented by a symbolic priority function. To ensure the applicability of the proposed method to various types of real-world traffic intersections, the priority function operates at the traffic movement level (Gu et al., 2024; 2025). It processes the traffic features linked to a specific traffic movement and generates a corresponding priority value. Figure 1 presents an example of SymLight's phase decision-making process at a classic intersection. A traffic signal phase is associated with several traffic movements (e.g., six in this figure). For each traffic movement, the priority function firstly takes its associated traffic features as inputs, and subsequently outputs the priority value of the traffic movement. Secondly, the priority values of traffic movements corresponding to each phase are summed, resulting in a distinct priority for each phase. Finally, the phase exhibiting the highest priority is ultimately selected as the next phase, as it is considered the most urgent. Consequently, its corresponding traffic lights will turn green.

#### 3.1.1 SCALE TO MULTIPLE INTERSECTIONS

Employing agents with shared parameters (Chen et al., 2020; Wei et al., 2019b; Liao et al., 2024; 2025a) for managing various intersections is demonstrated to be highly efficient and effective re-

cently. Rather than learning numerous intersection-specific priority functions, all intersections utilize a shared priority function to control their traffic lights.

### 3.1.2 Traffic features

As the priority function operates at the traffic movement granularity, it exclusively considers features from the two lanes (incoming and outgoing) directly associated with a specific movement. SymLight's priority function takes eight traffic features as input, which are obtainable with ease via road sensors.

- The number of waiting vehicles on the incoming (F1=WI) and outgoing (F2=WO) lanes.
- The count of vehicles on the incoming (F3=CI) and outgoing (F4=CO) lanes.
- The count of vehicles present on the incoming (F5=DI) and outgoing (F6=DO) lanes within a specific range of their respective approaching intersection. This range is defined as the distance vehicles can traverse through the intersection during a single phase's green light duration.
- The occupancy ratio for the incoming lane (F7=LI), calculated as CI divided by the total capacity of the incoming and outgoing lanes; and similarly for the outgoing lane (F8=LO), derived from CO.

Notably, for normalization, all features except LI and LO (which are already within [0,1]) are divided by the total number of vehicles in the intersection, bringing all values into the [0,1] range. The priority function will be constructed using the previously mentioned traffic features as variables, along with basic operators such as addition, negation, multiplication, protected division (return 1 if the denominator equals 0), minimum, and maximum.

### 3.2 Priority Function Representation

The priority function $\pi$ for SymLight is constituted by operators and variables. The task in this work features a relatively higher number of variables compared to traditional symbolic regression tasks. Therefore, a well-designed MCTS expansion rule for the MCTS state (i.e., the priority function), is crucial for preventing combinatorial explosion of the action space dimensionality.

In this work, we propose a simple yet effective representation of the priority function using a linear data structure that contains a list of operators and variables, such as $[+, -, \times, \text{WO}, \text{WI}, \text{WI}]$. We define an arity[1] attribute for each token $\tau$ in the list, such that operators have an arity greater than 0, and variables have an arity of 0. In this case, each variable is regarded as a nullary function. The ordered list representation for priority function is a breadth-first traversal (BFT) of its corresponding expression tree. Based on the BFT and the arity of each token within it, an expression tree can be uniquely constructed, thereby yielding its specific mathematical expression.

### 3.3 Monte Carlo Tree Search

Monte Carlo Tree Search (MCTS), an algorithm for finding optimal solutions in expansive discrete spaces represented by search trees, has been widely deployed. It has proven spectacularly successful in various gaming artificial intelligence systems, such as the well-known AlphaGo and AlphaZero (Silver et al., 2017; Fawzi et al., 2022). To provide context for the detailed steps of our MCTS, we begin by describing its components using the framework of a Markov Decision Process (MDP) with deterministic transitions (Shah et al., 2020). Notably, In SymLight, MCTS is not used for decision-time planning. Instead, it serves as an offline symbolic policy search algorithm that learns a fixed, interpretable priority function. During deployment, the obtained symbolic policy is executed directly without any online search.

#### 3.3.1 State

The MCTS state is represented by the aforementioned token list, which defines the priority function. Consequently, each node within the MCTS search tree corresponds to a candidate priority function.

---

[1]Arity is the number of arguments taken by a function.

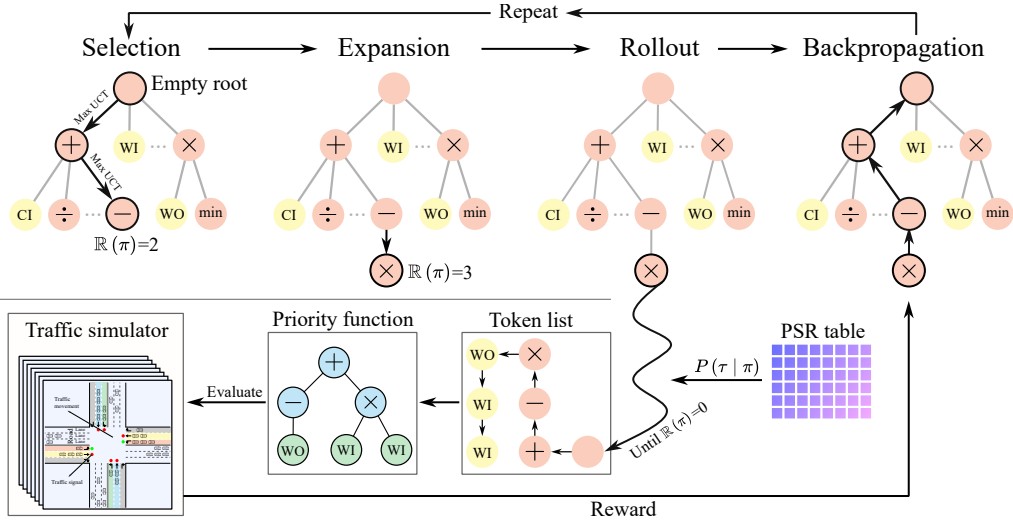

Figure 2: The overall framework of SymLight for exploring priority functions. To facilitate illustration, the state of an MCTS node is represented by the token list comprising all nodes on the path from the root to that node.

### 3.3.2 ACTION

MCTS nodes are expanded by incorporating a new token into their current state. Therefore, each individual token, either an operator or a variable, is considered as an MCTS action.

### 3.3.3 REWARD

The TSC problem's objective is to minimize the average travel time of all vehicles within a limited time frame (Gu et al., 2024; Wei et al., 2021). SymLight directly employs its global and ultimate objective as the reward. For a terminal node in MCTS, the reciprocal of the average travel time obtained from its priority function will serve as the node's reward.

MCTS involves a repetitive execution of four core steps (see Figure 2). Through this iterative process, the search is ultimately guided to discover promising priority functions for TSC.

1. **Selection**. During the selection step of MCTS, the goal is to follow a path within the search tree[2], commencing at its root, i.e. a empty list, and concluding at an expandable node or a terminal node, while balancing exploration and exploitation based on a selection strategy, i.e. the Upper Confidence Bounds applied for Trees (UCT) (Kocsis & Szepesvári, 2006). The selection strategy aims to maximize the Upper Confidence Bounds applied for Trees (UCT) (Kocsis & Szepesvári, 2006), which is mathematically expressed as follows:

$$UCT(\pi, \tau) = Q(\pi, \tau) + c\sqrt{\ln[N(\pi)]/N(\pi, \tau)}, \qquad (1)$$

   where $Q(\pi, \tau)$ signifies the estimated reward for taking action $\tau$ in state $\pi$. $N(\pi)$ indicates the total number of visits to state $\pi$; $N(\pi, \tau)$ tracks how often action $\tau$ was selected when in state $\pi$, and $c$ is a constant controls the balance between exploration and exploitation.

2. **Expansion**. Once the selection step reaches an expandable node of the search tree, an unvisited child of the node is expanded by randomly picking an action.

3. **Rollout**. After expansion, if the newly expanded node is not a terminal state, the algorithm then conducts random rollout, which begin at this node and continue until a terminal state is reached. The terminal state, acting as the priority function, is then evaluated to yield a reward $r$.

4. **Backpropagation**. Following the rollout step, the reward $r$ is backpropagated from the evaluated node to the root node. Since SymLight aim to find the priority function with

---

[2]Not to be confused with the tree-based expression of the priority function.

the best performance, the estimated reward $Q(\pi, \tau)$ is defined as the maximum reward among the sub-branches of this state-action pair. Therefore, for each involved node during the backpropagation, its estimated reward is updated as $Q(\pi, \tau) \longleftarrow \max(Q(\pi, \tau), r)$. Meanwhile, to overcome the local minima problems due to this greedy strategy, we adopt the $\epsilon$-greedy mechanism (commonly used in RL community) in the selection step.

### 3.3.4 VALIDATION FOR PRIORITY FUNCTION

The priority function representation is intentionally designed for flexibility and looseness, free from intricate syntactic rules, thereby facilitating easy validation. An ordered token list $\pi$ can be legitimately transformed into a priority function, if the following equation holds:

$$\sum_{\tau \in \pi} \operatorname{arity}(\tau) + 1 = |\pi|, \tag{2}$$

where $|\pi|$ means the list length. The above formula enables us to construct a criterion for verifying whether an arbitrary token list can be properly translated into a valid mathematical function:

$$\mathbb{R}(\pi) = 1 + \sum_{\tau \in \pi} \operatorname{arity}(\tau) - |\pi|, \tag{3}$$

where the value of $\mathbb{R}(\pi)$ indicates the remaining number of tokens (with arity of 0) necessary to complete the current token list into a legitimate priority function. When $\mathbb{R}(\pi) = 0$, $\pi$ is a valid priority function. Therefore, for any MCTS node with state $\pi$, $\mathbb{R}(\pi) = 0$ signifies a terminal node; otherwise, it is an expandable node.

### 3.3.5 EXPANSION RULE

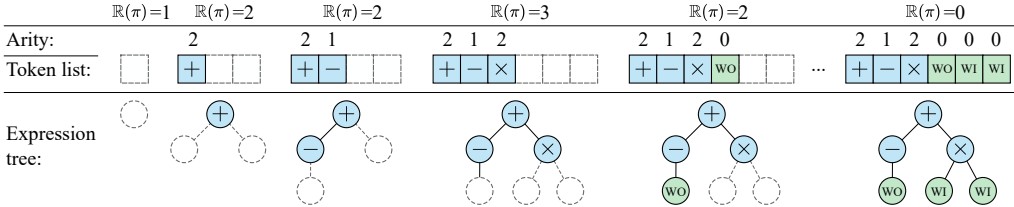

Figure 3: Example of priority function expansion. The priority function undergoes a sequential random expansion, proceeding from left to right. This process continues until $\mathbb{R}(\pi) = 0$, when its expansion is finalized, culminating in the mathematical expression $\pi(\cdot) = \text{WI} \times \text{WI} - \text{WO}$.

The flexible nature of the priority function representation imposes no strict constraints on expansion, as newly added tokens can be either operators or variables. Starting with an empty list, a complex priority function can be built by incrementally adding a single token (uniformly sampled from a combined set of operators and variables) to the end of the list. An example of priority function expansion is illustrated in Figure 3. To curb the unbounded expansion of the priority function, we impose a limit on the maximum number of operators. Once the current priority function's operator count surpasses a defined threshold, subsequent expanded tokens can only be variables. Due to the straightforward expansion rule of the priority function, the number of actions at each MCTS node is upper-bounded by the sum of operators and variables.

### 3.3.6 PROBABILISTIC STRUCTURAL ROLLOUT (PSR) STRATEGY

The rollout step in MCTS typically involves selecting actions uniformly at random until a terminal state is reached. However, this often generates low-quality sub-structure for priority function. To address this limitation, we propose the Probabilistic Structural Rollout (PSR) strategy, which leverages structural patterns observed in previously discovered high-quality priority functions to guide the rollout process more effectively.

The core idea of PSR is to replace uninformed random rollout with a structure-aware, experience-guided sampling process. This is achieved by maintaining a PSR table $c(p, \tau)$ that models the

frequency of each parent-child pair, based on the structural statistics collected from promising expression trees of priority functions. This table is constructed incrementally based on the historically $k$-best priority functions encountered during the MCTS search. For each expression tree of the high-quality priority function, we traverse its structure. For every observed parent-child pair in the tree, we simply increment the count in the corresponding table cell.

Based on the PSR table, a conditional probability distribution $P(\tau \mid \pi)$ can be computed for use in rollout:

$$P(\tau \mid \pi) = \frac{c(p, \tau) + \alpha}{\sum_{\tau'} c(p, \tau') + \alpha |\mathcal{A}(p)|}, \tag{4}$$

where $p$, a token already in $\pi$, represents the parent for next token of $\pi$; and $\alpha$ is the initial value of the PSR table, serving as a smoothing factor to avoid zero probabilities and retain some exploration. In this work, we simply use fixed default values of $\alpha = 1$ and $k = 10$ across all datasets. Our parameter sensitivity analysis (details in the supplementary material) shows that these values provided consistently strong performance, without dataset-specific tuning.

### 3.3.7 ADAPTIVE REWARD SHAPING

The reward's range, which is tied to the average vehicle travel time, is both unknown and dependent on the dataset. To address this, we adopt an adaptive reward shaping strategy. In this strategy, all raw rewards are normalized by the current best reward observed during the search. This normalization ensures that all reward values lie in the range $(0, 1]$, allowing the MCTS algorithm to maintain consistent and stable $Q(\pi, \tau)$ estimates without requiring a dataset-dependent exploration constant $c$ in UCB (Sun et al., 2023). Therefore, the exploration constant is set to a fixed value $\sqrt{2}$ across all datasets.

## 4 EXPERIMENTS

### 4.1 EXPERIMENTAL SETUP

We evaluate SymLight on six real-world traffic datasets provided by the CityFlow simulator, including networks from Hangzhou, Los Angeles, Atlanta, Jinan, and Manhattan, following the settings in prior works (Gu et al., 2024; Xing et al., 2022). We compare SymLight against conventional (Max-Pressure(Varaiya, 2013)), DRL-based (MPLight(Chen et al., 2020), CoLight(Wei et al., 2019b), FRAP(Zheng et al., 2019a)), practical (EcoLight(Chauhan et al., 2020), TinyLight(Xing et al., 2022), GPLight+(Liao et al., 2025a), $\pi$-Light(Gu et al., 2024)), and symbolic-regression–adapted baselines(Sun et al., 2023; Landajuela et al., 2021; Petersen et al., 2019), using average travel time and throughput as evaluation metrics. More details about experimental settings are provided in the Appendix.

### 4.2 PERFORMANCE

The performance of each method on different scenarios is summarized in Table 1. The table clearly indicates that traditional methods consistently show weaker performance than their learning-based counterparts. DRL-based methods tend to have larger variance than practical methods. The proposed SymLight demonstrates the superior performance in terms of average travel time among all the compared baselines in all the scenarios. Also, SymLight consistently exhibits the highest throughput across five distinct scenarios. The only exception is the Atlanta scenario, where SymLight exhibits only a small gap from the best compared algorithm, $\pi$-Light. Additionally, the adapted SPL and DSO algorithms, operating within the SymLight framework, surpass existing practical methods, further demonstrating the benefits of the proposed SymLight framework.

### 4.3 INTERPRETABILITY

The primary advantages of symbolic, explicit TSC policies are their transparency and human-understandability. Figure 4a presents two priority functions derived from Hangzhou1 datasets by SymLight. Their mathematical expressions are $\pi_1(\cdot) = LI \times DI^2$ and $\pi_2(\cdot) = LI \times \min(DI, DI^2)$. Given $0 \leq DI \leq 1, DI^2 \leq DI$. Consequently, the simplified expression for

| | Hangzhou1 | | Hangzhou2 | | Los Angeles | |
|---|---|---|---|---|---|---|
| | Travel Time (sec./veh.) | Throughput (veh./min.) | Travel Time (sec./veh.) | Throughput (veh./min.) | Travel Time (sec./veh.) | Throughput (veh./min.) |
| MaxPressure | $121.23_{\pm3.07}$↑ | $31.19_{\pm0.09}$ | $138.72_{\pm1.75}$↑ | $23.10_{\pm0.05}$ | $588.24_{\pm22.79}$↑ | $23.84_{\pm1.57}$ |
| CoLight | - | - | - | - | - | - |
| FRAP | $129.65_{\pm45.77}$↑ | $30.79_{\pm1.21}$ | $137.02_{\pm34.42}$↑ | $23.09_{\pm0.37}$ | $719.70_{\pm94.50}$↑ | $12.70_{\pm7.86}$ |
| MPLight | $119.58_{\pm43.37}$↑ | $30.96_{\pm0.96}$ | $122.55_{\pm1.75}$↑ | $23.24_{\pm0.06}$ | $577.96_{\pm81.35}$↑ | $23.63_{\pm6.96}$ |
| EcoLight | $189.52_{\pm9.56}$↑ | $29.95_{\pm0.24}$ | $135.41_{\pm2.21}$↑ | $23.16_{\pm0.24}$ | $650.52_{\pm11.07}$↑ | $19.98_{\pm0.73}$ |
| TinyLight | $102.87_{\pm2.98}$↑ | $31.36_{\pm0.05}$ | $121.00_{\pm0.85}$↑ | $23.25_{\pm0.04}$ | $489.93_{\pm19.76}$↑ | $31.29_{\pm2.73}$ |
| GPLight+ | $91.93_{\pm1.60}$↑ | $31.49_{\pm0.04}$ | $119.35_{\pm1.23}$≈ | $23.25_{\pm0.05}$ | - | - |
| $\pi$-Light | $92.31_{\pm1.16}$↑ | $31.52_{\pm0.06}$ | $119.15_{\pm1.09}$≈ | $23.26_{\pm0.05}$ | $446.09_{\pm29.45}$↑ | $34.59_{\pm3.90}$ |
| SymLight-SPL | $92.04_{\pm1.05}$↑ | $31.48_{\pm0.06}$ | $119.31_{\pm0.94}$≈ | $23.25_{\pm0.03}$ | $429.95_{\pm14.48}$↑ | $36.32_{\pm0.83}$ |
| SymLight-DSO | $91.05_{\pm1.01}$≈ | $31.50_{\pm0.04}$ | $119.02_{\pm0.92}$≈ | $23.26_{\pm0.03}$ | $414.33_{\pm11.76}$≈ | $37.91_{\pm0.67}$ |
| SymLight | $\mathbf{90.14_{\pm1.46}}$ | $31.52_{\pm0.06}$ | $\mathbf{118.80_{\pm0.92}}$ | $\mathbf{23.26_{\pm0.03}}$ | $\mathbf{410.28_{\pm10.48}}$ | $\mathbf{38.21_{\pm0.91}}$ |

| | Atlanta | | Jinan | | Manhattan | |
|---|---|---|---|---|---|---|
| | Travel Time (sec./veh.) | Throughput (veh./min.) | Travel Time (sec./veh.) | Throughput (veh./min.) | Travel Time (sec./veh.) | Throughput (veh./min.) |
| MaxPressure | $261.01_{\pm4.20}$↑ | $59.44_{\pm1.51}$ | $340.13_{\pm1.69}$↑ | $94.76_{\pm0.19}$ | $315.83_{\pm10.99}$↑ | $43.09_{\pm0.14}$ |
| CoLight | - | - | $856.53_{\pm451.32}$↑ | $57.96_{\pm30.08}$ | $1713.36_{\pm33.92}$↑ | $2.20_{\pm0.72}$ |
| FRAP | $258.21_{\pm18.27}$↑ | $63.49_{\pm8.19}$ | $327.90_{\pm23.28}$↑ | $95.10_{\pm1.17}$ | $531.90_{\pm361.58}$↑ | $33.26_{\pm12.48}$ |
| MPLight | $248.48_{\pm9.88}$↑ | $67.03_{\pm4.41}$ | $295.23_{\pm4.13}$↑ | $96.37_{\pm0.13}$ | $195.61_{\pm4.76}$↑ | $44.92_{\pm0.10}$ |
| EcoLight | $303.07_{\pm14.93}$↑ | $47.63_{\pm5.26}$ | $384.43_{\pm9.64}$↑ | $92.08_{\pm0.79}$ | $1014.61_{\pm33.11}$↑ | $17.65_{\pm0.90}$ |
| TinyLight | $253.99_{\pm8.08}$↑ | $62.16_{\pm3.77}$ | $310.62_{\pm3.68}$↑ | $95.79_{\pm0.39}$ | $322.01_{\pm168.10}$↑ | $40.55_{\pm6.55}$ |
| GPLight+ | - | - | $280.99_{\pm0.72}$↑ | $96.94_{\pm0.16}$ | $197.25_{\pm1.56}$↑ | $44.90_{\pm0.06}$ |
| $\pi$-Light | $233.16_{\pm1.90}$↑ | $\mathbf{70.51_{\pm0.37}}$ | $275.40_{\pm0.48}$↑ | $97.07_{\pm0.13}$ | $205.35_{\pm0.45}$↑ | $44.77_{\pm0.07}$ |
| SymLight-SPL | $232.22_{\pm2.78}$≈ | $69.64_{\pm0.78}$ | $275.86_{\pm0.60}$↑ | $97.03_{\pm0.11}$ | $194.77_{\pm2.52}$↑ | $44.93_{\pm0.07}$ |
| SymLight-DSO | $231.69_{\pm2.27}$≈ | $70.40_{\pm0.66}$ | $275.59_{\pm0.43}$↑ | $97.03_{\pm0.14}$ | $193.04_{\pm0.46}$↑ | $44.98_{\pm0.08}$ |
| SymLight | $\mathbf{231.07_{\pm2.24}}$ | $69.98_{\pm0.86}$ | $\mathbf{274.71_{\pm0.48}}$ | $\mathbf{97.08_{\pm0.05}}$ | $\mathbf{191.28_{\pm0.29}}$ | $\mathbf{44.99_{\pm0.04}}$ |

Table 1: The performance of different algorithms on six real-world road networks (sec. = second, veh. = vehicle, min. = minute). The symbols "↑/≈" indicate whether SymLight is significantly better than or statistically comparable to the corresponding baseline, based on the Wilcoxon rank-sum test at a significance level of 0.05 with Bonferroni correction. The symbol "-" means the corresponding baseline is not compatible with heterogeneous intersections that have different phase configurations. CoLight with intersection communication is not suitable for Hangzhou1 and Hangzhou2.

$\pi_2$ equals $\pi_1$. While expressive neural networks can, in theory, represent simple symbolic policies, their complexity potentially hinder them from reliably learning such straightforward policies. Contrary to directly employing end-to-end neural policies, symbolic policies can achieve excellent performance when integrated with a meticulously crafted decision-making framework. The explicit nature of the derived policies by SymLight reveals the direct relationship between each traffic feature and phase priority. For instance, the example priority function in Figure 4a highlights that a greater number of vehicles in the incoming lane closer to the intersection ($DI^2$) leads to a higher

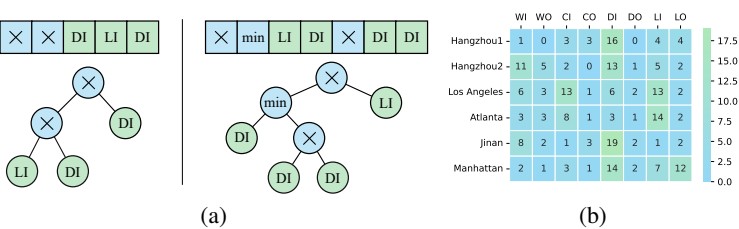

Figure 4: (a) Two example priority functions obtained in Hangzhou1 datasets. (b) The occurrence frequency of each traffic feature within the optimal solutions across six scenarios.

priority for the phase associated with that lane. The transparent nature of TSC policies ensures they are readily understandable, examinable by experts, and thus foster user trust.

Another advantage of using transparent TSC policies is gaining valuable insight into the relative importance of traffic features. To understand the importance of various traffic features, Figure 4b shows the occurrence frequency of each feature within the best learned policies across different scenarios. From the figure, it can be observed that traffic feature importance is heterogeneous across different scenarios. Overall, features associated with incoming lanes are typically more crucial, given that traffic light adjustments have the most direct influence on vehicles in these lanes.

### 4.3.1 ADDITIONAL ANALYSES

To further evaluate our method, we conduct four additional analyses. *Robustness*: SymLight exhibits insensitivity to variations in its hyperparameters (e.g., $\alpha$ and $k$ in the PSR strategy). *Ablation Study*: Each component of SymLight—reward shaping, enhanced traffic features, and the PSR strategy—contributes meaningfully to performance. *Deployability*: The TSC policies generated by SymLight are efficient in terms of computation time and memory usage, and are compatible with commonly used traffic signal controllers. *Generalization*: SymLight generalizes well to unseen scenarios without retraining and consistently outperforms baseline methods. Due to space constraints, detailed results are provided in the Appendix.

## 5 CONCLUSIONS

In this paper, we propose SymLight, a symbolic TSC policy search framework based on MCTS. In SymLight, the TSC policy functions based on a symbolic priority functions with a newly proposed representation, and MCTS is employed to search for optimal priority functions in the non-differentiable heuristic space. The experiment results on multiple real-world datasets demonstrate that each component of SymLight is effective based on the ablation study, and SymLight outperforms all the baseline methods. Further analysis of the priority functions generated by SymLight reveals that our method yields TSC policies that are human-interpretable, verifiable to domain experts, deployable on resource-constrained edge devices, and generalizable to novel scenarios.

Future work includes extending SymLight to multi-objective settings that jointly consider metrics such as delay, fairness, and emissions, exploring user-equilibrium and system-optimal formulations for network-level coordination, enriching the traffic state with additional traffic features, and designing improved search strategies that more effectively balance exploration and exploitation.

### REPRODUCIBILITY STATEMENT

All algorithmic details (cf. Section 3), hyperparameters, and implementation specifics (cf. Section A.5) are documented in the main paper and further elaborated in the Appendix. For empirical studies, we relied on publicly available datasets (cf. Section A.5) and applied the Wilcoxon rank-sum test with a significance level of 0.05 to account for randomness (cf. Section 4). Upon acceptance, we will release our code to facilitate reproduction of our experiments. Together, these resources ensure that independent researchers can replicate our results and extend our contributions.

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

# A    APPENDIX

| | Hangzhou1→Hangzhou2 | | Hangzhou2→Atlanta | | Hangzhou2→Los Angeles | |
|---|---|---|---|---|---|---|
| | Travel Time (sec./veh.) | Throughput (veh./min.) | Travel Time (sec./veh.) | Throughput (veh./min.) | Travel Time (sec./veh.) | Throughput (veh./min.) |
| MaxPressure | $135.89_{\pm 1.94}$ | $23.23_{\pm 0.03}$ | $261.01_{\pm 4.20}$ | $59.44_{\pm 1.51}$ | $588.24_{\pm 22.79}$ | $23.84_{\pm 1.57}$ |
| FRAP | $173.75_{\pm 146.11}$ | $22.48_{\pm 2.25}$ | $362.41_{\pm 79.07}$ | $26.23_{\pm 26.38}$ | $860.70_{\pm 29.59}$ | $2.73_{\pm 2.75}$ |
| MPLight | $145.37_{\pm 73.13}$ | $22.91_{\pm 0.94}$ | $328.49_{\pm 72.26}$ | $40.22_{\pm 25.63}$ | $865.16_{\pm 12.61}$ | $2.30_{\pm 0.95}$ |
| EcoLight | $134.60_{\pm 0.92}$ | $23.15_{\pm 0.03}$ | $386.03_{\pm 52.96}$ | $18.53_{\pm 17.55}$ | $739.19_{\pm 81.49}$ | $11.66_{\pm 7.48}$ |
| TinyLight | $703.15_{\pm 316.77}$ | $14.27_{\pm 4.54}$ | - | - | - | - |
| GPLight+ | $120.06_{\pm 1.01}$ | $23.25_{\pm 0.03}$ | - | - | - | - |
| $\pi$-Light | $119.95_{\pm 1.54}$ | $23.26_{\pm 0.03}$ | $242.55_{\pm 4.64}$ | $66.25_{\pm 2.08}$ | $487.31_{\pm 44.96}$ | $28.14_{\pm 6.43}$ |
| SymLight-SPL | $120.33_{\pm 0.83}$ | $23.25_{\pm 0.04}$ | $247.97_{\pm 4.12}$ | $62.56_{\pm 2.31}$ | $489.13_{\pm 41.38}$ | $28.31_{\pm 6.59}$ |
| SymLight-DSO | $121.19_{\pm 1.14}$ | $23.25_{\pm 0.04}$ | $243.81_{\pm 2.17}$ | $66.10_{\pm 1.62}$ | $465.48_{\pm 43.70}$ | $32.17_{\pm 6.04}$ |
| SymLight | $\mathbf{119.91}_{\pm 1.15}$ | $\mathbf{23.26}_{\pm 0.05}$ | $\mathbf{240.14}_{\pm 4.23}$ | $\mathbf{67.35}_{\pm 1.91}$ | $\mathbf{461.83}_{\pm 33.96}$ | $\mathbf{33.42}_{\pm 4.69}$ |

Table A2: The performance of different algorithms on six real-world road networks (sec. = second, veh. = vehicle, min. = minute). "-" means some methods cannot be transferred due to different intersection phase configuration.

| Parameter | Symbol | Value |
|---|---|---|
| Initial value for PSR table | $\alpha$ | 1.0 |
| Set size of priority functions | $k$ | 10 |
| Decision interval | $\Delta t$ | 20 s |
| Training episodes | — | 500 |
| Maximum number of operators | — | 6 |
| Exploration constant of UCT | $c$ | $1/\sqrt{2}$ |
| $\epsilon$-greedy | $\epsilon$ | 0.2 |

Table A3: Parameter Settings for SymLight

## A.1    ABLATION STUDY

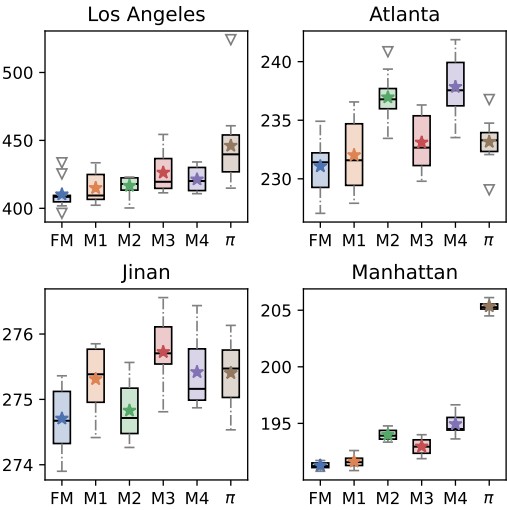

Figure A5: The ablation study results in terms of average travel time, where lower values indicate better performance. (FM = SymLight, $\pi$ = $\pi$-Light)

To confirm the individual contribution of each component within SymLight, we performed four separate ablation studies. Each study involved either systematically removing or modifying a key

component, and then analyzing its impact on overall performance. Specifically, we evaluate the average travel time of vehicles on multiple scenarios under several settings:

1. **M1** The adaptive reward shaping mechanism is removed, and the reciprocal of the average travel time is directly employed as the reward.

2. **M2** The newly introduced features, LI and LO, are removed from the input of the priority functions.

3. **M3** The PSR strategy is replaced with the default random rollout in MCTS.

4. **M4** All three modifications from M1, M2, and M3 are applied simultaneously.

Notably, SymLight is labeled as the full model (FM). The results of the ablation study are illustrated in Figure A5. The results of the state-of-the-art, $\pi$-Light, are also included as a reference, and shown as $\pi$ in the figures. It is evident from the results that every component contributes positively. While these components show modest improvements in some scenarios, they consistently deliver substantial performance gains in others, affirming the value of each individual part. Interestingly, we observe that M4 occasionally outperforms M3. This suggests that the PSR strategy (removed in M3) is particularly important for robust performance across various scenarios, underscoring the necessity and importance of this component in the overall model design.

## A.2 MODEL GENERALIZATION

To validate the generalization ability of the proposed SymLight, TSC polices are trained in a source environment, and then directly evaluated in a target environment without any retraining. As shown in Table A2, DRL-based methods have exhibited poor generalization performance, suggesting that their highly expressive and complex neural policies potentially suffer from overfitting. Conversely, EcoLight shows only minor performance degradation, which can possibly be attributed to its simple network architecture reducing the likelihood of overfitting. SymLight-based methods and $\pi$-Light achieve superior generalization due to their highly concise policy representations. Additionally, SymLight consistently outperforms other baselines when generalizing to unseen environments, maintaining the best performance.

## A.3 PARAMETER SENSITIVITY ANALYSIS

In this section, we analyze the robustness of SymLight with respect to its hyperparameters and the fixed phase duration. Specifically, we examine the impact of (1) the initial value $\alpha$ of the PSR table, which serves as a smoothing factor, (2) the number $k$ of top-performing priority functions used to construct the PSR table, and (3) the phase duration. These experiments aim to verify the insensitivity of SymLight to different configurations, demonstrating that strong performance can be maintained without careful hyperparameter tuning.

We test $\alpha$ values from the set $\{0.5, 0.8, 1.0, 1.2, 1.5\}$. As a smoothing factor, $\alpha$ mitigates zero probabilities for unseen structures in early rollout steps and maintains exploration. As shown in Figure A6, SymLight achieves consistently strong performance across all tested values. The differences in performance are minimal, demonstrating that the model is insensitive to the choice of $\alpha$. We also evaluate the influence of the set size $k \in \{5, 8, 10, 12, 15\}$, which determines how many high-quality priority functions are utilized for PSR table construction. As shown in Figure A6, SymLight performs robustly across this range, with no significant degradation observed. This suggests that the method does not heavily rely on precise tuning of $k$. To test responsiveness under different signal control granularities, we vary the phase duration across $\{10s, 15s, 20s\}$. Figure A7 shows that SymLight maintains high-quality traffic signal control performance across all settings.

Overall, the results indicate that SymLight is not sensitive to the values of $\alpha$ and $k$, and it continues to perform effectively under different phase durations. These findings demonstrate that SymLight can be deployed across various traffic scenarios without extensive hyperparameter tuning, enhancing its practicality and robustness in real-world applications.

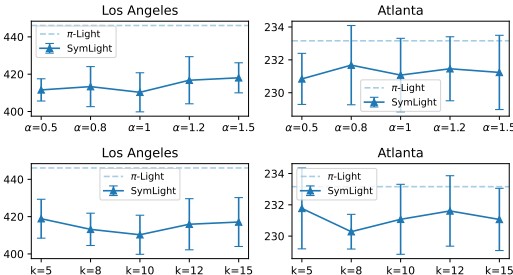

Figure A6: The performance (average travel time in seconds, the smaller the better) sensitivity of SymLight to different hyperparameter ($\alpha$ and $k$) settings.

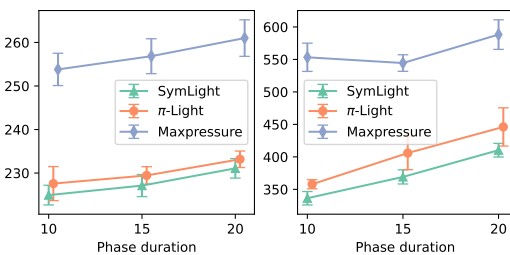

Figure A7: Performance (average travel time in seconds, the smaller the better) comparison under phase durations of 10, 15, and 20 seconds, respectively

### A.4 DEPLOYABILITY

A significant benefit of symbolic policies over neural policies is their ability to be deployed on resource-limited edge devices. We analyze each algorithm's TSC policy model based on two criteria: storage and computation. These two factors collectively dictate the minimum device configuration needed (Xing et al., 2022). To gauge computation time and memory consumption, we utilize floating-point operations (FLOPs) and model parameter size (bytes), respectively.

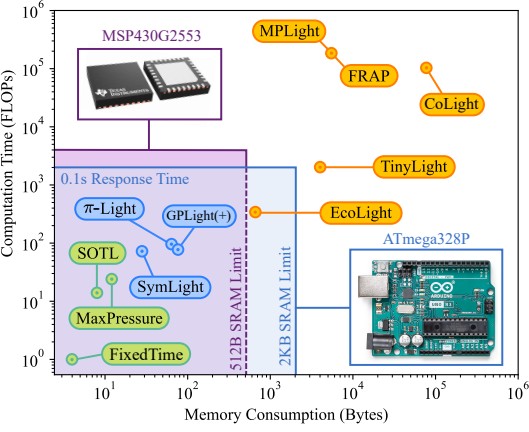

Figure A8: Log-log plot for resource consumption of each TSC policy from different methods.

Figure A8 presents the FLOPs and parameter sizes of each model. The figure clearly shows that neural policies derived from DRL-based methods demand significant computation time and memory consumption. While TinyLight and EcoLight use concise neural network structures, significantly reducing resource demands, they still have a substantial gap when compared to conventionally deployable methods. With its highly constrained resource consumption, the proposed SymLight

closely approximates traditional approaches. Notably, SymLight's symbolic policies also show a slight resource advantage over $\pi$-Light's program-form policies. Additionally, to provide a more intuitive understanding of these values, the figure displays the RAM limits and 0.1-second response time thresholds of two commonly used (Jyothi et al., 2016) microcontrollers, ATmega328P[3] and MSP430G2553[4], assuming an average of 400 cycles per operation on the target microcontroller[5]. As observed, the SymLight-evolved policies are deployable on these resource-limited edge devices, suggesting that our approach offers potential advantages in real-world traffic deployment.

## A.5 EXPERIMENT SETUP

### A.5.1 DATASETS AND SIMULATOR

The experiments are performed using CityFlow, an open-source traffic simulator tailored for handling traffic signal control scenarios. Following previous works (Gu et al., 2024), we use six traffic datasets collected from real-world cities, including Hangzhou1 (road network of 1×1), Hangzhou2 (1×1), Los Angeles (4×1), Atlanta (5×1), Jinan (4×1), and Manhattan (16×3). Notably, intersections in the road networks of Los Angeles and Atlanta are heterogeneous, featuring different shapes and diverse phase configurations.

The datasets record each vehicle's entry time into the road network and its route. In order to quantify the robustness of the diverse methods to noise, we generate nine additional traffic flows for each road network within every environment. These were obtained by displacing the entry time of each vehicle in the real-world traffic records with random noise values ranging from -60 to 60.

All the experiments were conducted on a high-performance computing cluster equipped with dual Intel Xeon E5-2695v4 processors, utilizing the Slurm workload manager for job scheduling. All baselines are trained with sufficient number of episodes until convergence. As per convention (Xing et al., 2022), each green signal is followed by a three-second all-red interval to clear vehicles at intersections. The minimum action duration was set to 20 seconds, maintaining uniformity throughout all baseline methods. All parameters of SymLight are specified in the supplementary materials.

### A.5.2 BASELINES

We establish our experimental baselines using a diverse set of methods, categorized as follows:

- **Conventional Transportation Methods** We include MaxPressure (Varaiya, 2013) to represent traditional TSC approaches.

- **DRL-based Methods** This category comprises three state-of-the-art DRL-based methods MPLight (Chen et al., 2020), CoLight (Wei et al., 2019b), and FRAP (Zheng et al., 2019a) that do not consider real-world deployment.

- **Practical Methods** We also incorporate EcoLight (Chauhan et al., 2020), TinyLight (Xing et al., 2022), GPLight+ (Liao et al., 2025a), and $\pi$-Light (Gu et al., 2024), specifically designed with practical deployment constraints in mind.

- **SymLight-Adapted Symbolic Regression methods** Additionally, we adapt two state-of-the-art symbolic regression methods Symbolic Physics Learner (SPL) (Sun et al., 2023) and Deep Symbolic Optimization (DSO) (Landajuela et al., 2021; Petersen et al., 2019) to fit the proposed SymLight framework for comparative analysis.

We employ two prevalent evaluation metrics (Xing et al., 2022; Chen et al., 2020; Liao et al., 2025a): Travel Time, representing the average vehicle travel duration on the network, and Throughput, indicating the number of vehicles passing per minute. Optimally, Travel Time should be minimized, and Throughput maximized.

---

[3]https://store.arduino.cc/products/arduino-uno-rev3/

[4]https://www.ti.com/product/MSP430G2553

[5]For processors without a hardware FPU, this value is typically 300–500 cycles per operation

### A.5.3 PARAMETER SETTING

The parameter settings for SymLight are summarized in Table A3. All the experiments were conducted on a high-performance computing cluster equipped with dual Intel Xeon E5-2695v4 processors, utilizing the Slurm workload manager for job scheduling. The algorithms were implemented in Python, and the simulator was implemented in C++, running on a CentOS operating system. Additionally, the simulation time (episode length) varies by environment, as it is dependent on the duration of the traffic flow records. The duration of each episode is set to 900 seconds for Atlanta, 1800 seconds for Los Angeles, and 3600 seconds for all remaining environments.

## A.6 TRAINING EFFICIENCY

To provide a clear picture of the computational footprint associated with different methods, we report in Table A4 the number of traffic simulation episodes and the corresponding wall-clock time (in minutes) required for training on four scenarios that are representative in terms of scale.

Table A4: Training episodes and wall-clock time (in minutes) for each method.

| Method | Episodes | Hangzhou1$_{1\times1}$ | Los Angeles$_{4\times1}$ | Atlanta$_{5\times1}$ | Manhattan$_{16\times3}$ |
|---|---|---|---|---|---|
| MPLight | 128 | 7.28m | 65.09m | 7.74m | 302.21m |
| $\pi$-Light | 500 | 3.29m | 14.59m | 4.55m | 121.28m |
| GPLight+ | 500 | 0.78m | - | - | 65.74m |
| SymLight | 500 | 3.51m | 19.80m | 3.82m | 218.93m |

All baselines are trained with a sufficient number of episodes until convergence.
Different TSC policies lead to varying traffic conditions within the network, resulting in different simulation runtimes.
Training time is also influenced by implementation-specific details of each codebase.

Although SymLight requires more simulation episodes than MPLight, its overall training time is much shorter than that of MPLight. The explanation is as follows. In MPLight, both forward inference and backpropagation are required at every training step, introducing additional computational overhead beyond the cost of simulation. In contrast, SymLight maintains constant-time operations for both node expansion and rollout due to its symbolic function space and lightweight tree structure. As a result, the computational bottleneck for SymLight lies almost entirely in traffic simulation rather than tree search operations, making its symbolic optimization process efficient and stable across scenarios. Furthermore, GPLight+ exhibits a significantly reduced offline training time compared to other algorithms. We hypothesize that this efficiency stems from the population-based search mechanism, which effectively restricts the exploration and evaluation of policies to a relatively favorable solution landscape. This minimizes the possibility of discovering extremely poor policies, which in turn leads to a substantial reduction (the simulation runtime can is influenced by the specific TSC policy) in the required simulation time.

## A.7 ANALYSIS OF THE DISCOVERED PRIORITY FUNCTIONS

To better illustrate the interpretability of the symbolic priority functions discovered by SymLight, we present representative examples obtained from each scenario. Table A5 shows the token lists produced by MCTS and their corresponding simplified symbolic expressions. These concise algebraic forms allow traffic engineers to easily inspect, validate, and reason about the behavior of the learned policies.

The symbolic expressions in this table reveal meaningful patterns that align with established traffic engineering principles. Across scenarios, demand-related features such as DI and WI frequently play dominant roles, reflecting an intuitive demand-driven logic where phases with heavier incoming traffic receive higher priority. Moreover, scenario-specific structures naturally emerge, such as the emphasis on $DI^2$ in Hangzhou1, which captures strong upstream demand, or the use of $-\min(LI, LO)$ in Atlanta, which accounts for downstream congestion and matches the known congestion propagation characteristics of that network. Recurrent motifs including $DI + WI$, $DI^2$, and $\max(\cdot)$ further demonstrate that MCTS combined with PSR consistently identifies stable and meaningful symbolic structures across heterogeneous intersections. These examples collectively

Table A5: Representative priority functions discovered in different scenarios.

| Scenario | Token List | Simplified Expression |
|---|---|---|
| Hangzhou1 | $[\times, \times, DI, LI, DI]$ | $DI^2 \times LI$ |
| Hangzhou2 | $[\max, +, LI, \max, DI, WI, WI]$ | $\max(LI, DI + WI)$ |
| Los Angeles | $[\times, \times, LI, +, LI, LI, WI]$ | $LI^2 \times (LI + WI)$ |
| Atlanta | $[+, -, LI, +, \min, WO, LI, LO]$ | $LI - WO - \min(LI, LO)$ |
| Jinan | $[+, \times, DI, DI, WI]$ | $DI \times (WI + 1)$ |
| Manhattan | $[\max, \max, DI, DI, \times, LO, LO]$ | $\max(DI, LO^2)$ |

show that the symbolic policies produced by SymLight are concise and well aligned with real-world operational logic.

Figure A9 presents the Pareto front between the complexity (the number of operators) of the learned priority functions and the resulting system performance across all evaluated scenarios. The system performance is measured based on the average travel time (ATT). Consequently, superior performance corresponds to a lower ATT value. Each curve illustrates how increasing the number of operators within the symbolic priority function affects the performance. As shown in this figure, the performance initially improves as complexity grows, but quickly saturates once a modest level of expressiveness is reached. Beyond this threshold, additional complexity yields negligible performance gains, indicating that concise symbolic policies remain highly effective. These results provide a clear basis for practitioners (e.g., transportation agencies) to balance performance against policy complexity according to their operational needs, and they highlight SymLight's capability to generate compact yet high-performing control strategies.

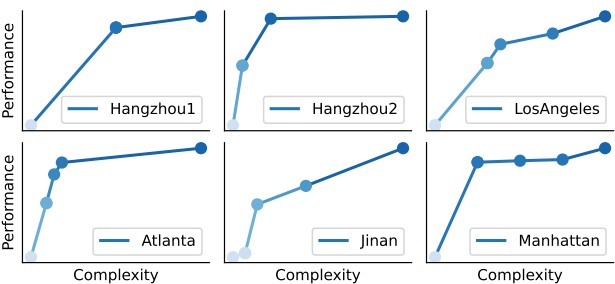

Figure A9: The Pareto Front of performance versus complexity for the priority functions derived by SymLight on each scenario.

## A.8 ROBUSTNESS TO SENSOR NOISE

In real-world traffic environments, sensor measurements are often subject to noise due to imperfect detection, communication delays, or environmental conditions. To evaluate the robustness of SymLight to such uncertainties, we conduct a Gaussian noise attack on the input traffic features of the priority function. Specifically, for each feature, we add an independent Gaussian noise $z \sim N(0, \sigma^2)$, where $\sigma^2$ controls the noise level.

Figure A10 shows the performance of the learned priority functions under increasing levels of noise. As expected, the average travel time gradually increases with higher $\sigma^2$, indicating a performance degradation. Importantly, even under relatively high noise levels, SymLight maintains an acceptable performance that surpasses MaxPressure, demonstrating that the discovered symbolic priority functions are robust and reliable against realistic sensor uncertainties.

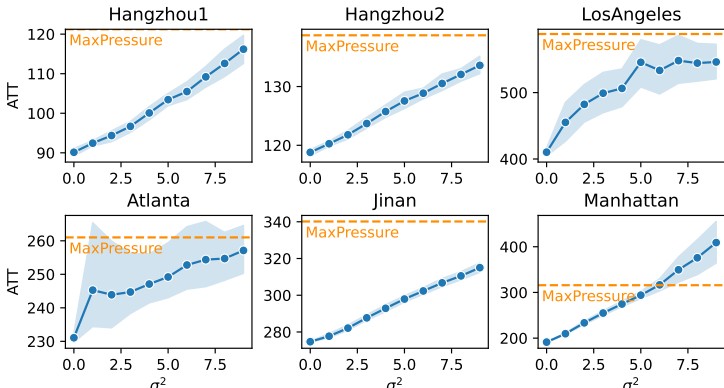

Figure A10: Priority function performance as the Gaussian noise $z \sim N(0, \sigma^2)$ in traffic features increases. Under noise levels where the scale $\sigma^2 < 5$, the performance of SymLight is not inferior to the noiseless MaxPressure.

## LLM USAGE STATEMENT

We employed ChatGPT (GPT-5) solely as an assistive tool for grammar checking and language refinement. The model was not used for research ideation, algorithm design, experiment execution, or result analysis. All scientific content and conclusions are entirely the work of the authors.

