# OpenReview forum: "SymLight: Exploring Interpretable and Deployable Symbolic Policies for Traffic Signal Control"
_ICLR.cc/2026/Conference — Submitted to ICLR 2026_

### Official Review · Reviewer_mGfV · 2025-10-20

**Soundness:** 3
**Presentation:** 3
**Contribution:** 3
**Rating:** 6
**Confidence:** 5

**Summary:**

This paper proposed a Monte Carlo Tree Search method to improve the interpretability of traffic signal control. Experiments show that the performance is achieved a new SOTA.

**Strengths:**

Using Monte Carlo Tree Search to improve the interpretability of traffic signal control sounds interesting. Experiments show that the performance is achieved a new SOTA. This method is very simple and effective.

**Weaknesses:**

1. Why use repeat WI in [+, −, ×, WO, WI, WI]? This example makes reader confused.

2. More newly baseline methods should be compared. More scalability datasets should be experimented.

3. How is the robustness of your method? In real-world deployment, there may be some noise in perception, could the method still keep high performance[1,2]?

4. If we add running vehicles as input, can the performance be improved?

5. This performance could be getting stuck in local optima due to greedy selection of the best reward in current step, it ignores long-term reward payoffs. The authors should discuss this issue and address how to resolve it as future work.


[1]Robustlight: improving robustness via diffusion reinforcement learning for traffic signal control. ICML
[2]Fuzzylight: a robust two-stage fuzzy approach for traffic signal control works in real cities

**Questions:**

See weakness.

---

> ### Author Response · Authors · 2025-11-26
>
> We sincerely thank you for the positive and constructive comments/suggestions, which are very helpful for improving our paper. Please find our responses below. Revisions have also been made in the paper.
>
> ## Q1 Why WI Appears Twice
>
> Thanks for your comments.
> Thank you for pointing out the confusion regarding the example. We clarify that:
>
> 1. SymLight **does not restrict the number of times a traffic feature can appear** in a symbolic expression. Instead, the search space explicitly allows features to appear multiple times so that the resulting symbolic functions can represent richer algebraic structures.
>
> 2. In the given example, the repeated WI tokens are not redundant. They form a meaningful term. Specifically, the two WI tokens are combined by the multiplication operator, yielding a squared feature term $WI^2$. The full expression expands to:
> $$WI^2-WO.$$
> Thus, the observed repetition is not mere redundancy, but rather reflects a deliberate and meaningful structure discovered by the MCTS search process.
>
> ## Q2 More Baselines And Scalability Datasets
>
> Thanks for your comments.
> In this work, we compare SymLight against a diverse set of baselines, including: (i) conventional TSC methods (MaxPressure), (ii) representative DRL methods (MPLight, CoLight, FRAP), (iii) practical deployment-oriented methods (EcoLight, TinyLight, GPLight+, $\pi$-Light), and (iv) SymLight-adapted symbolic regression methods (SPL, DSO). This coverage ensures both performance and practicality are evaluated comprehensively.
> Following your suggestion, we are currently conducting comparisons with extra baselines.
> Due to computational requirements, these experiments are ongoing and the results will be presented in the final version.
>
> Regarding scalability, our experiments span networks of varying size and complexity, from medium-scale (Atlanta, Los Angeles, Jinan) to large-scale (Manhattan), demonstrating that SymLight maintains high performance and efficiency even as network scale grows. We are additionally running experiments on further large-scale networks to reinforce these findings.
>
>
> ## Q3 Robustness to Perception Noise in Real-World Deployment
>
> Thank you for raising this important concern.
> We have conducted additional experiments, applying Gaussian noise attacks directly to all traffic features used by the priority function.
> The results show that SymLight experiences a gradual performance degradation as the noise magnitude increases, while still maintaining acceptable performance within the tested range. More details about the results can be found in Sec. A.8 of the Appendix.
>
> Additionally, we have also included and discussed about `RobustLight` and `FuzzyLight` in the related work section as both are important works in practical TSC methods.
>
> > Recently, some researchers begin to explore practical TSC methods for real-world deployment.
> Recent works such as `RobustLight` (Li et al., 2025b) and `FuzzyLight` (Li et al., 2025a) have made impressive progress in enhancing the robustness of TSC policies under perception noise and uncertain traffic conditions. Their innovative approaches demonstrate that TSC systems can maintain high performance even in extremely challenging real-world scenarios.
> In addition to this, ...
>
> ## Q4 Using Additional Inputs for Potential Performance Gain
>
> Yes, prior work has demonstrated that incorporating fine-grained vehicle-level features can significantly improve performance. For example, Advanced Traffic State (Zhang et al., ICML 2022) introduces "Effective Running Vehicles" as part of the state representation and reports notable improvements across DRL-based TSC frameworks.
> Such features can be seamlessly plugged into our symbolic function representation, and we believe this presents a promising avenue for extension, especially in future work involving high-quality sensing environments.
>
> However, to prioritize deployability on existing traffic infrastructure, the use of these challenging vehicle-level traffic features is deferred from the scope of the present work.

---

> ### Author Response · Authors · 2025-11-26
>
> ## Q5 Possible Local Optima from Greedy Search
>
> We appreciate your insightful comment regarding potential local optima.
>
> We would like to clarify that **SymLight can directly optimize the long-term reward rather than the immediate reward at current step**. This is achieved by the following designs. First, unlike traditional MCTS, where each node's Q-value represents the expected reward (average returns), SymLight aims to discover the single best priority function rather than maximizing average returns over all substructures. Therefore, during Backpropagation, we only propagate the maximum reward observed along the current branch.
>
> Second, for node selection, SymLight employs the **UCT selection strategy**, which balances exploration and exploitation, and we further adopt the commonly used **$\epsilon$-greedy mechanism** to mitigate the risk of being trapped in local optima.
>
> Last but not least, **SymLight does not suffer from the temporal credit assignment problem**. SymLight directly optimizes the given system-level objective, such as average travel time, instead of relying on surrogate objectives (e.g., intersection queue length, intersection pressure) that DRL methods often use to indirectly improve real-world metrics (Salimans, Tim, et al., 2017).
>
> We hope this response would help clarify the your concern. Please do not hesitate to let us know if you have any further questions. Thank you very much.

---

> > ### Comment · Reviewer_mGfV · 2025-11-27
> >
> > I appreciate the hard work of author. The one thing that i have concerned is after reading about the paper PI-Light, i found the MCTS method have been proposed by other paper, could the author discuss about the novelty or clarify the difference of your method?

---

> > > ### Author Response · Authors · 2025-11-27
> > >
> > > Thank you for your reply.
> > > We would like to clarify that the novelty of our method over $\pi$-Light.
> > >
> > > **1. Policy Representation**
> > >
> > > SymLight utilizes non-parametric, symbolic algebraic functions that are continuous, purely mathematical, and fundamentally unrestricted in form. This contrasts with $\pi$-Light, which employs DSL-based programs with learnable parameters, making its expressiveness and flexibility inherently constrained by the DSL design and its explainability potentially detrimentally impacted by the parameters. In summary, our representation is more compact, which enables the optimizer to conduct a more effective global search for optimal policies.
> > >
> > > **2. Efficient Expansion**
> > >
> > > We designed an effective encoding scheme for our policy representation, coupled with a highly efficient MCTS expansion rule. Compared to $\pi$-Light, our approach successfully mitigates the combinatorial explosion of the MCTS action space (the action space of SymLight scales only linearly with the number of tokens). Importantly, this design future-proofs our method, guaranteeing the scalability required for incorporating advanced and finer-grained traffic features (such as running vehicles) enabled by future sensor technology development.
> > >
> > > **3. Guided Search**
> > >
> > > Compared to $\pi$-Light, we have introduced key adjustments to the MCTS search mechanism. Specifically, we propose a probabilistic structural rollout strategy that leverages structural patterns extracted from high-quality expressions to guide efficient exploitation during the search process.
> > >
> > > The experimental results demonstrate that SymLight significantly (statistically) outperforms $\pi$-Light. Furthermore, the results from SymLight-SPL and SymLight-DSO also demonstrate the advantage of symbolic policies over programs policies.

---

### Official Review · Reviewer_ouBD · 2025-10-31

**Soundness:** 4
**Presentation:** 3
**Contribution:** 4
**Rating:** 8
**Confidence:** 4

**Summary:**

This paper develops a method for identifying a symbolic priority function for traffic intersection management. The approach uses MCTS to design interpretable, symbolic decision rules that score lanes for selection. Empirical results show that the method reduces travel time and increases vehicle throughput across 6 intersection scenarios. Through its use of a symbolic function, the resulting decision rule is automatically interpretable.

**Strengths:**

- The use of a symbolic priority function is a well-motivated approach for providing interpretability in an application domain where interpretability is a main bottleneck to realizing gains from machine learning.
- The paper presents its main ideas clearly.
- The empirical analysis supports the central claims of the paper: improved intersection metrics and interpretability of the learned priority function.

**Weaknesses:**

- Clarity and relevance to learning: it took me a couple of read-throughs of the method to understand how the priority function was implemented and what the role of MCTS was. I still give the paper a good rating for clarity because -- once I understood the method -- I think the organization makes sense. So this weakness is more about, "could the paper have been more clear?" My confusion was that MCTS is usually not a learning method itself but is a decision-time search method. So at first I expected the method to use MCTS to select the next phase. There then appeared to be no learning in the paper which would make it out of ICLR's scope. Ultimately, I see that MCTS is used **offline** to learn a fixed priority function. This is not a major weakness but I believe more clarity could be added in this regard.
- Table 1's font is difficult to read without zooming in a lot.
- It wasn't fully clear how datasets are being used. I am assuming that the data provides the demand profiles for each scenario so that you know how many vehicles to simulate?

**Questions:**

Please see weaknesses.

---

> ### Author Response · Authors · 2025-11-26
>
> We sincerely thank you for the positive and constructive comments/suggestions, which are very helpful for improving our paper. Please find our responses below. Revisions have also been made in the paper.
>
> ## Q1 Clarifying the Role of MCTS as an Offline Learning Procedure
>
> Thank you for pointing this out. We clarify that MCTS is indeed used purely offline as a learning mechanism to search for a symbolic priority function. After the offline search converges, the learned symbolic function is fixed and deployed for real-time control with no further search or online planning.
> This approach is beneficial as it prevents the need for continuous online optimization during actual deployment, thereby avoiding potential decision latency.
>
> To avoid confusion, we have explicitly stated this in the paper (Sec. 3.3):
>
> > Notably, In SymLight, MCTS is not used for decision-time planning. Instead, it serves as an offline symbolic policy search algorithm that learns a fixed, interpretable priority function. During deployment, the obtained symbolic policy is executed directly without any online search.
>
> ## Q2 Improving Table Readability
>
> Thanks for your comments.
> We have enlarged the font and reformat Table 1 to ensure readability in the final version.
>
> ## Q3 Datasets Clarification
>
> Thanks for your comments. Yes, that is correct, we can know how many vehicles to simulate.
> Moreover, we also know the **spatiotemporal distribution of traffic flow** and **topological and geometric information of the traffic network** that enables us to construct even more realistic simulations.
> The datasets provide real-world demand profiles and network configurations, including:
>
> * lane/road/intersection geometric structure
> * traffic signal phase configurations
> * traffic flow datasets recording each vehicle's entry time into the road network and its complete route (so the simulator can know its origin and destination)
>
> Based on the above data, the simulator can construct a traffic simulation that closely approximates real-world scenarios.
>
> We hope this response would help clarify the your concern. Please do not hesitate to let us know if you have any further questions. Thank you very much.

---

> > ### Comment · Reviewer_ouBD · 2025-11-26
> >
> > Thank you for clarifying these points.

---

### Official Review · Reviewer_XcGG · 2025-11-10

**Soundness:** 3
**Presentation:** 3
**Contribution:** 2
**Rating:** 4
**Confidence:** 4

**Summary:**

The paper introduces SymLight, a priority function search framework for traffic signal control (TSC) that balances interpretability, efficiency, and deployability. Unlike deep reinforcement learning (DRL) models, which are opaque and resource-intensive, SymLight discovers explicit symbolic priority functions through Monte Carlo Tree Search (MCTS) to decide traffic light phases based on real-time traffic features. It proposes a concise symbolic representation, an adaptive reward shaping mechanism, and a probabilistic structural rollout strategy that leverages structural patterns from high-quality expressions to guide efficient exploration. SymLight directly optimizes system-level objectives (e.g., average travel time) and produces lightweight, human-understandable control policies suitable for deployment on low-cost edge devices. Extensive experiments on six real-world traffic networks (Hangzhou, Los Angeles, Atlanta, Jinan, and Manhattan) show that SymLight outperforms state-of-the-art DRL and symbolic baselines in both travel time reduction and throughput improvement.

**Strengths:**

1. The paper proposes a symbolic-policy formulation for traffic signal control, representing control logic as explicit priority functions. This approach is promising to bridge the gap between high-performance learning methods and human-interpretable rule-based systems, addressing a long-standing limitation in DRL-based TSC.
2. By integrating MCTS with a concise symbolic representation and the proposed probabilistic structural rollout strategy, SymLight enables efficient exploration of large discrete spaces.
3. Experiments on six real-world traffic networks demonstrate consistent improvements in both travel time reduction and throughput over strong DRL and symbolic baselines.

**Weaknesses:**

1.	While the central idea of this work lies in leveraging a priority function search for traffic signal control, the paper provides limited background or theoretical motivation for this concept. As a result, the rationale for adopting priority functions and their novelty relative to existing symbolic or TSC works remains unclear.
2.	The Monte Carlo Tree Search has been explored in traffic signal optimization [1]. The authors are expected to clarify the difference and novelty compared to the existing literature.
3.	Can you intuitively explain how your model achieves better interpretability and deployability than existing methods?
4.	The code and implementation details are not publicly available, which hinders the reviewers to reproduce and verify the reported results.

[1] Monte Carlo Tree Search-based intersection signal optimization model with channelized section spillover. Transportation Research Part C: Emerging Technologies. 2019.

**Questions:**

Please refer to the Weaknesses.

---

> ### Author Response · Authors · 2025-11-26
>
> We sincerely thank you for the positive and constructive comments/suggestions, which are very helpful for improving our paper. Please find our responses below. Revisions have also been made in the paper.
>
> ## Q1 Motivation and Background for Priority Functions
>
> Thanks for the comments. The inspiration for adopting a heuristic (in this work the priority function serves as the heuristic) to calculate the priority of each phase and subsequently control the signal signals is drawn from **MaxPressure theory** (Varaiya, 2013). Following your comments, we have clarified this in the revised manuscript.
>
> In Sec. 2.2 of the manuscript, we described the limitations of existing methods. To enable readers to quickly grasp the key points, the summarization is presented in the following table:
>
> | Method Category | Policy Representation | Interpretability | Deployability | Notes |
> |-|-|-|-|-----------------------------------|
> | Traditional (Fixed-time, Actuated) | Hand-crafted rules | ✓ High | ✓ High | Unsatisfactory performance in dynamic traffic; dependent on expertise |
> | DRL-based Methods (FRAP, CoLight, MPLight) | Neural networks | ✗ Low | ✗ Low | Black-box policies; reward–objective mismatch; high deployment costs |
> | LLM-based TSC | Language models | ✓ | ✗ Hard to deploy | High decision cost; overly fixed prompt template limits generalization in heterogeneous scenarios |
> | Lightweight Neural Policies (TinyLight, EcoLight) | Compressed NN models | ✗ Low | ✓ Medium (still NN-based) | Performance drop due to compression; still opaque |
> | Symbolic / Program-based TSC (GPLight+, π-Light, DRSQ) | Polynomials, parameterized programs | ✓ High | ✓ High | Inflexible policy representation or incompatible with heterogeneous settings |
> | SymLight (Ours) | Non-parametric mathematical expressions | ✓ Explicit | ✓ High | Discover human-understandable, deployable, and non-parametric TSC policies, without compromising performance |
>
> We would like to clarify that the core novelty of our approach, SymLight, over existing methods is threefold:
>
> 1. **Novel Policy Representation** We introduce a new policy representation in the form of a non-parametric, symbolic priority function. This function directly maps the real-time traffic features of each phase to a single priority value, which subsequently governs the traffic signal control decision.
>
> 2. **Encoding Scheme** We designe a concise encoding scheme for the priority function. This scheme effectively mitigates the combinatorial explosion issue within the MCTS action space during tree expansion. Specifically, the action dimension in SymLight scales linearly with the number of tokens.
>
> 3. **Guided Search** We propose a probabilistic structural rollout strategy that leverages structural patterns extracted from high-quality expressions to guide efficient exploitation during the search process.
>
> Numerical experiments demonstrate that SymLight can successfully discover human-understandable and deployable TSC policies while simultaneously maintaining excellent performance.
>
> ## Q2 Differences from MCTS-IO
>
> Thanks for the comments.
> The MCTS-IO method proposed in the reference also utilizes MCTS for TSC, leveraging MCTS to perform online optimization of the phase sequence during runtime. This approach is fundamentally distinct from our work, where MCTS is used as a search tool for symbolic policy generation in an offline fashion. The detailed conceptual comparison with MCTS-IO is shown below.
>
> |MCTS-IO|SymLight|notes|
> |-|-|-|
> |online optimization|offline search|Performing online optimization during decision-making stage can lead to potential decision latency|
> |non-interpretable phase sequence|human-understandable symbolic policy|The output of MCTS-IO is an immediate sequence of actions, which inherently lacks transparency. Conversely, SymLight's policy is represented as an algebraic expression (the priority function), which possesses inherent interpretability|
> |traffic flow model-dependent| traffic flow model-agnostic | The rollout phase of MCTS-IO is heavily dependent on a pre-constructed macro traffic flow model (used to simulate complex dynamics like CSS). In sharp contrast, SymLight achieves its goals by interacting directly with the simulation environment, thereby remaining independent of any underlying traffic dynamics model|
> |sequence search|policy search|MCTS-IO searches for the optimal sequence of actions (to be executed immediately). In contrast, SymLight searches for the optimal TSC policy (a generalized function)|
>
> Following your comments, we have included the referenced work and discussed about its difference from our approach in the related work section of the revised manuscript.

---

> ### Author Response · Authors · 2025-11-26
>
> ## Q3 Intuitive Interpretability and Deployability
>
> Thanks for the comments. Below are the explanation on why our model can achieve better interpretability and deployability.
>
> **Interpretability**:
>
> In SymLight, the learned symbolic function is a short algebraic expression involving human-understandable traffic features, e.g. queue length on the incoming lane (WI) and outgoing lane (WO).
> $$\mathbb{P}(s) = \sum_{\text{TM}} WI-WO.$$
>
> In contrast to neural policies, which consist of numerous matrices of abstract numerical values, the symbolic expressions are composed of traffic features that carry specific physical meaning.
> This composition makes them readable at a glance and crucially allows traffic engineers to easily verify, understand, and trust the policy behavior.
>
> **Deployability**:
>
> The symbolic policy of SymLight:
>
> * Decisions require only **a few simple arithmetic operations**, unlike neural policies that depend on large-scale floating-point computation and nonlinear activations.
> * Symbolic policies can be executed directly (including C or even assembly implementations) on existing **low-cost edge controllers without GPUs or hardware upgrades**, whereas neural policies often require model distillation or device replacement.
> * Deployment is not solely a technical or hardware issue. It also involves socio-technical considerations. The discovered symbolic policies are concise and transparent, the behavior of policies can be reliably predicted. This crucial transparency helps mitigate safety concerns raised by relevant authorities during practical deployment. In sharp contrast, the inherent black-box nature of neural policies often makes relevant stakeholders **hesitant to assume accountability for their behavior**, particularly in safety-critical systems.
>
> ## Q4 Code Availability
>
> Thank you for raising the concern about code availability.
> We would like to clarify that the implementation of SymLight has already been provided in the Supplementary Material during submission.
> Upon acceptance, we will release the complete version of the codebase and documentation as stated in the paper.
>
>
> We hope this response would help clarify the your concern. Please do not hesitate to let us know if you have any further questions. Thank you very much.

---

### Official Review · Reviewer_CMMS · 2025-11-10

**Soundness:** 3
**Presentation:** 3
**Contribution:** 2
**Rating:** 4
**Confidence:** 5

**Summary:**

The paper introduces SymLight, a traffic-signal control method that searches for symbolic priority functions—compact algebraic expressions mapping lane-level features to phase priorities—via MCTS. The priority function is encoded as a token list over basic operators (addition, negation, multiplication, min/max, protected division) and lane features. The search adds a Probabilistic Structural Rollout (PSR) that biases rollouts using parent-child token statistics from the top-k expressions.  The reward is adaptively normalized by the current best observed travel time to stabilize UCT.  Experiments on six CityFlow road networks (Hangzhou, Los Angeles, Atlanta, Jinan, Manhattan) report lower average travel time and higher throughput than conventional.

**Strengths:**

1. Interpretable policy class with a easy understand over meaningful lane features; protected division and min/max are practical choices.

2. Empirical improvements on six CityFlow networks with significance testing.

3. Search framework is simple to implement; PSR is a plausible way to avoid uninformative rollouts.

**Weaknesses:**

1. The core claim is that SymLight yields strong, deployable policies; however, the offline search costs (simulation calls, expansions, rollouts, wall-clock per network/intersection) are not quantified. Without this, it’s unclear whether the approach scales to larger grids or frequent retuning.

2. The reward normalizes inverse travel time, but multi-objective considerations (emissions, per-approach fairness, pedestrian delay) are absent, and it is unclear whether the method over-optimizes one metric at others’ expense.

3.  Regarding the interpretability, more analysis case should added. The robustness of model interpretability is need to be considered.

4. More symbolic RL works need to be compared, which can help hightlight the contribution of this work.

Discovering symbolic policies with deep reinforcement learning, ICML, 2021.

Learning Neurosymbolic Generative Models via Program Synthesis, 2019, ICML.

Neurosymbolic Reinforcement Learning with Formally Verified Exploration, 2020, ICML.

5.  What are the per-network wall-clock hours, simulator steps, node expansions, and rollout counts for MCTS+PSR to reach the reported policies? Please tabulate alongside DRL training costs.

6. How does performance vary with k and \alpha? Any evidence of PSR biasing search toward substructures that generalize poorly across intersections?

**Questions:**

Please see weakness.

---

> ### Author Response · Authors · 2025-11-26
>
> We sincerely thank you for the positive and constructive comments/suggestions, which are very helpful for improving our paper. Please find our responses below. Revisions have also been made in the paper.
>
> ## Q1&Q5 Offline Search Cost
>
> We agree that reporting the computational footprint is important for assessing deployability.
> The number of MCTS rollouts, node expansions are aligned because each node expansion corresponds to exactly one rollout in SymLight.
> The traffic simulation episodes and wall-clock minutes are tabled as follows:
>
> | Method | Episodes | Hangzhou1 | Los Angeles | Atlanta | Manhattan |
> |-|-|-|-|-|-|
> |MPLight|128| 7.28m | 65.09m |7.74m| 302.21m |
> |$\pi$-Light|500| 3.29m |14.59m| 4.55m | 121.28m |
> |GPLight+|500| 0.78m |-| - | 65.74m |
> |SymLight|500|3.51m| 19.80m | 3.82m | 218.93m |
>
> Notably:
>
> 1. All baselines are trained with sufficient number of episodes until convergence
>
> 2. Different TSC policies lead to varying traffic conditions within the road network, which consequently results in different simulation runtimes
>
> 3. Training time is also influenced by the specific implementation of the code.
>
> Compared to MPLight, SymLight requires more simulation episodes.
> However, due to the need for backpropagation and inference time in MPLight, the total time consumption of DRL is actually longer than that of SymLight.
> In SymLight, both node expansion and rollout are operations with constant complexity (thanks to the limited number of functions and the proposed concise representation). The computational bottleneck is therefore only the traffic simulation rather than tree search operations.
>
> Additionally, in real-world traffic scenarios that require real-time control, the efficiency of decision-making is particularly crucial. Compared to neural policies, symbolic policies offer extremely high decision efficiency, which is very important for practical deployment.
> The table below lists the comparison of decision/inference time between symbolic policies and neural policies. As can be seen, our method (SymLight) exhibits a much smaller decision time, which is beneficial for scaling to large-scale traffic scenarios.
>
> ||SymLight| MPLight |CoLight|
> |-|-|-|-|
> |Time for 100 Decisions|4.9ms|187.3ms|98.1ms|
>
> ## Q2 Multi-Objective Concerns
>
> Our method optimizes average travel time (ATT) because it is the primary metric used in previous TSC works. ATT was also be demonstrated as an effective comprehensive traffic metric (Wei, Hua, et al., SIGKDD, 2019). The smaller its value, the better the other traffic indicators usually are. Our paper also reports the results for both the ATT and vehicle throughput. These results demonstrate that SymLight, despite being primarily optimized for ATT, is capable of achieving the most competitive vehicle throughput across the majority of the tested scenarios.
>
> To validate that SymLight does not over-optimize a single metric to the detriment of others, we include results for two additional crucial traffic indicators: Average Delay (AD) and Average Queue Length (AQL) (both derived from the CityFlow API). These metrics are presented in the two subsequent tables.
>
> AD($\frac{1-v_{\text{vehicle}}}{v_{\text{limit}}}$, the smaller the better):
>
> |Method|Hangzhou1|Hangzhou2|Los Angeles|Atlanta|Jinan|Manhattan|
> |-|-|-|-|-|-|-|
> |$\pi$-Light|0.4766|0.4514|0.3546|0.6075|0.1887|0.0491|
> |GPLight+|0.4802|0.4527|-|-|0.1896|0.0421|
> |SymLight|**0.4756**|**0.4473**|**0.3508**|**0.5434**|**0.1833**|**0.0397**|
>
> AQL(the smaller the better):
>
> |Method|Hangzhou1|Hangzhou2|Los Angeles|Atlanta|Jinan|Manhattan|
> |-|-|-|-|-|-|-|
> |$\pi$-Light|1.0650|1.6011|0.8670|**2.9178**|0.4383|0.0691|
> |GPLight+|1.0458|1.6158|-|-|0.4887|0.0544|
> |SymLight|**0.9956**|**1.5918**|**0.8002**|2.9777|**0.4272**|**0.0489**|
>
>
> Although we acknowledge that definitively ruling out all possibilities of over-optimization across all scenarios requires extensive study, the inclusion of AD and AQL results demonstrates that our method does not severely degrade performance on key, unoptimized metrics.
>
> Additionally, the optimization of symbolic policies in SymLight does not require tricky reward design. It can be conveniently used for multi-objective optimization to find the Pareto front. This characteristic further demonstrates SymLight's advantages.
>
> Lastly, your suggestion of per-approach fairness is indeed a crucial metric, as it relates to the complex and challenging topic of balancing System Optimum with User Equilibrium in the traffic area. We are thankful for this insight and commit to including a discussion of this as a major focus for our future research directions.

---

> ### Author Response · Authors · 2025-11-26
>
> ## Q3 Interpretability Analysis and Robustness
>
>  Following your suggestions, we have added extra analysis case (see the following table) in sec. A.7 of the appendix.
>
> | Scenario | Token list | Expression(simplified) |
> |-|-|-|
> | Hangzhou1 | [$\times$, $\times$, DI, LI, DI] | $DI^2\times LI$ |
> | Hangzhou2 | [$\max$, +, LI, $\max$, DI, WI, WI]  | $\max(LI, DI+WI)$ |
> | Los Angeles | [$\times$, $\times$, LI, +, LI, LI, WI]  | $LI^2\times(LI+WI)$ |
> | Atlanta | [+, -, LI, +, $\min$, WO, LI, LO] | $LI-WO-\min(LI, LO)$ |
> | Jinan | [+, $\times$, DI, DI, WI] | $DI\times(WI + 1)$ |
> | Manhattan | [$\max$, $\max$, DI, DI, $\times$, LO, LO]  | $\max(DI, LO^2)$ |
>
> It can be observed that many discovered priority functions are relatively concise and align well with common sense (i.e., are intuitive). Specifically, phases associate to lanes with a higher volume of upstream vehicles are often designated as more urgent for a green light.
>
> To evaluate the robustness of the model's interpretability, we computed tree n-grams across the discovered priority functions. The top-5 sub-tree structure account for approximately 37% and the top-10 sub-tree structure account for approximately 56% of all occurrences. This suggest that the model maintains robust local structural patterns which supporting the stability of the model interpretability.
>
> Moreover, the inherent capability of SymLight for multi-objective optimization allows us to conveniently generate the Pareto front between priority function complexity and system performance. This provides users with a clear basis for making an explicit trade-off between model interpretability and operational efficacy.
> Crucially, our analysis reveals that beyond an acceptable complexity threshold for the priority function, the performance gains achieved by further increasing complexity (reduce interpretability) become negligible (all Pareto front curves have been provided in the Appendix). Consequently, SymLight demonstrates significant potential for discovering high-performance, yet highly concise, symbolic policies.
>
> ## Q4 Comparison with Neurosymbolic RL Works
>
> Thanks for the suggestion. We agree that positioning our work within symbolic/neurosymbolic RL is important.
> We have already included "Discovering symbolic policies with deep reinforcement learning, ICML, 2021." as a baseline in the original paper, and we will highlight this more clearly.
> Specifically, we incorporated its underlying algorithm into the SymLight paradigm, and it is explicitly referred to as SymLight-DSO within the paper.
>
> The methodologies employed in the second paper (focused on image generation) and the third paper (focused on action verification) exhibit a significant functional gap when compared to the TSC task.
>
> Given this substantial difference in domains and objectives, we find that a direct experimental comparison is highly challenging and may not be meaningful. Therefore, we have added a discussion on distinctions between their approaches and our method within the Related Work section of the revised paper.
>
> ## Q6 Sensitivity to Hyperparameters in PSR, and Potential Bias Issues
>
> We thank the reviewer for the question.
> Section A.3 of the Appendix already reports ablation experiments on both $k$ and $\alpha$.
> The results show that SymLight is not sensitive to the values of $k$ and $\alpha$.
>
> Verifying PSR biasing search toward substructures that generalize poorly across intersections directly is challenging.
> Therefore, we performed a zero-shot generalization test comparing SymLight with/without PSR.
>
> |  | Hangzhou1 $\rightarrow$ Hangzhou2 | Hangzhou2 $\rightarrow$ Atlanta | Hangzhou2 $\rightarrow$ Los Angeles |
> |-|-|-|-|
> | SymLight wo. PSR | 120.24±1.13 | 244.83±6.78 | 514.44±53.29 |
> | SymLight         | **119.91**±1.15 | **240.14**±4.23 | **461.83**±33.96 |
>
> The results indicate that the inclusion of PSR does not compromise the model's ability to generate solutions with poor generalization performance.
>
>
> We hope this response would help clarify the your concern. Please do not hesitate to let us know if you have any further questions. Thank you very much.

---

### Author Response · Authors · 2025-12-02
**General Response (1/3)**

We sincerely thank all reviewers for their comprehensive reviews and constructive feedback, which have significantly helped us improve the quality of our manuscript. We also appreciate the tremendous efforts of the newly assigned Area Chair in carefully reviewing our rebuttal and revised manuscript.

In the following overall summary, we provide a concise overview of the paper’s core contributions, summarizes the reviewers’ feedback and main concerns, and highlights how each concern was addressed.

## Paper Summary

This paper introduces SymLight, a novel Monte Carlo Tree Search (MCTS) based framework designed to discover inherently interpretable and deployable symbolic priority functions to serve as Traffic Signal Control (TSC) policies. SymLight directly addresses the critical limitations of current neural policies, namely their lack of interpretability and difficulty in deployment on resource-limited edge devices.

SymLight's key contributions are threefold:

1. **Novel Policy Representation** We introduce a new policy representation in the form of a non-parametric, symbolic priority function. This function directly maps the real-time traffic features of each phase to a single priority value, which subsequently governs the traffic signal control decision.

2. **Encoding Scheme** We design a concise encoding scheme for the priority function. This scheme effectively mitigates the combinatorial explosion issue within the MCTS action space during tree expansion. Specifically, the action dimension in SymLight scales linearly with the number of tokens.

3. **Guided Search** We propose a probabilistic structural rollout strategy that leverages structural patterns extracted from high-quality expressions to guide efficient exploitation during the search process.

---

### Author Response · Authors · 2025-12-02
**General Response (2/3)**

## Review Summary

The paper received four overall recommendations, resulting in a positive average score of 5.5.
Strengths unanimously noted by reviewers include:

(1) The strong motivation and successful achievement of inherently interpretable policies based on meaningful traffic features (`CMMS`&`ouBD`).
(2) The search framework is well-designed and the PSR strategy is a plausible innovation (`CMMS`&`XcGG`).
(3) The method is very simple and effective; the framework is simple to implement; the main ideas are presented clearly (`CMMS`&`ouBD`&`mGfV`).
(4) Superior empirical performance (SOTA in ATT and throughput) with significance testing across six real-world CityFlow networks (`XcGG`&`mGfV`).

**Consensus**: Three reviewers (`CMMS`, `XcGG`, `mGfV`) rated the quality and presentation as good (3), while reviewer `ouBD` provided an excellent rating (4). The primary point of contention was the Contribution score (`ouBD`: 4;`mGfV`: 3; `CMMS`, `XcGG`: 2), which the rebuttal was specifically designed to address. Reviewer `ouBD`'s high score is a significant initial endorsement of the work's quality.

---

### Author Response · Authors · 2025-12-02
**General Response (3/3)**

## Rebuttal and Discussion Summary

The rebuttal successfully addressed all major concerns and we have also updated the manuscript along with the appendix accordingly.

1. **Search Cost & Scalability** (`CMMS`, `mGfV`). The rebuttal provided a table for detailed offline training time, showing that SymLight's training time is shorter than DRL due to the significantly faster inference time and no need for backpropagation.
Moreover, the rebuttal provided comparisons on the online inference time, demonstrating SymLight is 10x to 38x faster than neural policies (e.g., 4.9ms vs. 187.3ms), which justifies the efficiency and scalability of our method.

2. **Novelty & Contribution** (`XcGG`, `mGfV`). 1. The rebuttal explicitly clarified that MCTS is used for offline learning to obtain optimized priority functions for direct online deployment, rather than online decision-making (addressing reviewer `ouBD`'s confusion and reviewer `XcGG`'s comparison to MCTS-IO). 2. Novelty over $\pi$-Light: the rebuttal detailed the technical novelties in non-parametric compact representation, flexible expansion rule, linear-scaling action space, and the PSR strategy (addressing reviewer `mGfV`'s confusion).

3. **Overfitting to the Optimized Metric** (`CMMS`). The rebuttal provided additional analysis on the Average Delay and Average Queue Length, demonstrating promising performance on a wide range of performance metrics, showing that our method does not overfit to the specific optimized objective/metric. In addition, our method is objective-driven rather than reward-based, and it naturally supports multi-objective optimization to discover trade-off policies in terms multiple objectives/performance metrics.

4. **Interpretability Robustness** (`CMMS`). The rebuttal includes additional experiments, and the new results demonstrate that the symbolic policies maintain robust local structural patterns. Besides, the rebuttal provided additional analysis cases of the priority function as well as Pareto front curves in the revised manuscript, which showed that the performance gains achieved by further increasing complexity of priority functions become negligible. This demonstrates our method has significant potential for discovering high-performance, yet highly concise, symbolic policies.

5. **Sensor Noise Robustness** (`mGfV`). The rebuttal provided additional experiments and analysis on Gaussian noise attack, showing graceful degradation of our performance over noises in input features. Even under high-noise conditions, the performance remains acceptable, i.e., still better than the Maxpressure method, which is one of the SOTA manual policy.

6. **Comparison to Symbolic RL** (`CMMS`). The rebuttal clear clarified that in the origin manuscript, the SOTA Symbolic RL method (ICLR's SPL, ICML's DSO) have already been integrated into our experimental comparisons as baselines (SymLight-SPL and SymLight-DSO), confirming comprehensive comparison coverage.

7. **Code Availability** (`XcGG`). In the initial submission, the source code has already been uploaded as a compressed file in the supplementary materials for reviewers to inspect (addressing reviewer `XcGG`'s concern). We will make the code publicly available after the paper is accepted/published.

---

We again thank the Area Chairs and Reviewers for their time and effort. Finally, we sincerely hope the Area Chair will consider our contributions to the community, along with the detailed rebuttal responses and the improved revised manuscript to address all reviewer concerns.

---

### Meta-Review · Area_Chair_Lptu · 2026-01-03

**Summary:**

This paper proposes SymLight, an offline Monte Carlo Tree Search framework for discovering symbolic priority functions for traffic signal control. By searching over compact algebraic expressions and introducing a probabilistic structural rollout strategy, the method aims to balance interpretability, efficiency, and performance. Experiments on six CityFlow networks demonstrate improvements over several DRL and symbolic baselines in average travel time and throughput.

Reviewers broadly agree that interpretability and deployability are important challenges in traffic signal control, and that symbolic policies are a promising direction. The proposed representation is concise and human-readable, and the empirical results show consistent performance gains on the tested benchmarks. Several reviewers found the method simple, effective, and practically motivated, with particularly strong inference-time efficiency.

Despite these positive aspects, multiple substantive concerns remain after rebuttal:

1. Limited Novelty and Conceptual Advancement.
   While the authors clarified differences from prior MCTS-based and symbolic TSC methods, the core approach—using MCTS to search for interpretable symbolic policies—remains closely aligned with existing literature. The incremental contributions (encoding choices, PSR, adaptive reward normalization) do not clearly amount to a conceptual advance commensurate with ICLR’s expectations, especially given prior work on symbolic policy discovery and MCTS-based traffic control.

2. Scalability and Practicality of Offline Search.
   Although additional computational cost reporting was provided, the offline search remains expensive, particularly on large networks. The feasibility of scaling SymLight to city-scale deployments or adapting to non-stationary traffic conditions requiring frequent retraining is not convincingly demonstrated.

3. Limited Experimental Scope Relative to Claims.
   The evaluation is confined to a single simulator and a small set of networks. While these are standard benchmarks, they are insufficient to fully support broader claims about general deployability, robustness, and superiority over alternative learning paradigms.


In summary, the contribution is primarily incremental, and the remaining concerns regarding novelty, scalability, and experimental breadth prevent a positive recommendation at this time. Given the competitive bar at ICLR and the narrow audience of this submission, I recommend **rejection**, while encouraging the authors to further strengthen the conceptual positioning and empirical validation in a future submission.

**Reviewer Concerns:**

As above

**Reviewer Scores:**

I do not think Reviewers XcGG and CMMS would change their scores.

---

### Decision · Program_Chairs · 2026-01-26

Reject